# A Dynamic Tool to Describe Lamb Growth and Its Use as a Decision Support System

**DOI:** 10.3390/ani14152246

**Published:** 2024-08-02

**Authors:** Rafael Marzall Amaral, Marcelo Teixeira Rodrigues, Erica Beatriz Schultz, Cristiano Eduardo Rodrigues Reis

**Affiliations:** 1EARTH University, San José 4442-1000, Costa Rica; 2Department of Animal Science, Federal University of Viçosa, Viçosa 36570-000, Brazilerica.schultz@ufv.br (E.B.S.)

**Keywords:** energy efficiency, feedlot, growth curve, performance simulation, system dynamics

## Abstract

**Simple Summary:**

A dynamic model was created to determine the nutritional requirements of lambs, aiming to simulate and describe their growth, nutrient requirements, and body composition. The model inputs include body mass, standard final mass, age, and dietary energy concentration. The results showed that the model accurately predicts the final body mass based on the inputs, with a coefficient of determination of 0.89 and a mean bias of 0.656 kg. The utility of this model extends to agricultural practitioners and livestock managers, providing valuable insights into forecasting final body mass and estimating the duration required to attain specific weights. Moreover, it serves as an educational tool elucidating the intricate interplay between nutrition, growth dynamics, and body composition in lambs. Thus, its implementation stands to enhance decision-making processes within feedlot lamb production, contributing to more informed and efficient management practices.

**Abstract:**

A dynamic model has been developed to simulate aspects of feedlot lamb growth and body composition, including energy and protein requirements, growth rate, composition of gain, and body mass. Model inputs include initial body mass (kg), standard final mass (kg), age (days), and dietary energy concentration (Mcal·kg^−1^). The model was assessed as a decision support tool using a dataset of 564 individual measures of final body mass and diet energy. The simulations provide graphical and numerical descriptions of nutrient requirements, composition of gain, and estimates of animal performance over time. The model is accurate and precise, with a root mean squared error of 7.79% of the observed final body mass and a coefficient of determination of 0.89 when simulating the same variable. The model can be used as a reliable decision support tool to estimate final body mass and the days on feed required to reach a certain final mass with precision and accuracy. Moreover, the dynamic model can also serve as a learning tool to illustrate practical principles of animal nutrition, nutrient requirement relationships, and body composition changes. This model holds the potential to enhance livestock management practices and assist producers in making informed decisions about feedlot lamb production.

## 1. Introduction

Proper nutrient requirement determination and nutrition of farm animals is a basic statement of sustainable intensification. Ensuring proper nutrition is crucial for promoting the growth and development of farm animals. Additionally, the economic feasibility of animal production is closely linked to the cost of nutrition [1]. To meet the nutritional needs of the increasing global population and to ensure food security, sustainable intensification involves maximizing the efficiency of land and other resources while minimizing the negative impact on the environment [2]. This approach to agriculture is crucial to meet the growing demand for food without causing irreparable harm to the environment.

Nutrition, digestion, metabolism, and determination of nutritional requirements and animal growth are complex dynamic systems that involve the ingestion of feeds, digestion, absorption, transport of nutrients, intermediary metabolism, underlying anabolism and catabolism, and the excretion of unabsorbed nutrients and metabolites. Their dynamic nature is due to the processes being influenced by both time and feedback relations between their constituent elements. Understanding these processes is crucial for aiding management decisions made by producers and animal scientists. With this knowledge, it becomes possible to formulate highly precise diets that can meet the animals’ needs in different production systems, estimate the quality and quantity of feed, and determine the time needed to achieve the final product. By utilizing resources in a careful and technical manner, it is possible to improve animal performance while minimizing environmental damage, resulting in fewer greenhouse gas emissions and pollutants being released into the atmosphere, all while remaining economically viable.

In this sense, mathematical modeling in animal production plays a critical role in forecasting and optimizing the animal growth process. Dynamic models differ from static models in that they consider time-dependent variables and can simulate changes over time, making them more suitable for predicting the behavior of biological systems under varying conditions. Dynamic modelling has been conducted in different areas of animal production, like farm production and management policies [3], diet preference and selection [4], methane emission by livestock [5], ruminal microbiology [6], small ruminants [7] dairy cattle production [8], and cattle growth [9,10]. The models have been used as predictive or descriptive elements as part of decision support systems and learning tools. 

Several models have been developed to improve our understanding of ruminant nutrition and growth. Notable among these are the Cornell Net Carbohydrate and Protein System (CNCPS), the Small Ruminant Nutrition System (SRNS) [11], and the Ruminant Nutrition System (RNS) [12], as well as NASEM [13,14]. The CNCPS is a mechanistic model that predicts the nutrient requirements and feed utilization for cattle based on rumen fermentation, microbial growth, and animal physiology. The SRNS, which evolved from the CNCPS, is tailored for sheep and goats, incorporating specific adjustments for small ruminants. Despite its advancements, it faces challenges in accurately predicting nutrient requirements for different breeds and environmental conditions. These models, with a mechanistic structure, establish these relations and give a significant contribution to their understanding with better descriptions of the different nutritional processes. All these models are static; they support only punctual estimation related to the animals’ needs and their individual performances, without feedback on or continuity of information flow. The RNS represents an advancement by integrating additional submodels and stochastic modeling capabilities to account for variability in feed composition and animal responses. It offers more accurate predictions by considering a wider range of variables and interactions, yet it still requires refinement to address the complexities of ruminant nutrition fully. A summary of these models is described by Tedeschi [15]. Within this framework, this article introduces a dynamic model that integrates the equations of the CNCPS-S, SRNS, and RNS models to obtain the protein and energy requirements, body composition of gain, and a fuller understanding of animal growth. It accounts for cyclical changes over time in feedlot lambs, from weaning until maturity, using animal and diet characteristics as inputs. 

## 2. Methods

### 2.1. Model Framework

The model was constructed using the System Dynamics approach and parts of the mechanistic equations proposed in the CNCPS-S [16], SRNS [11], and RNS [12] to describe the nutritional requirements, composition of gain, and dynamics of growth over time. This model was constructed on a set of ordinary differential equations based on the causal relations between selected variables with the arrangement of the main components of dynamic simulation like stocks, flows, feedback, delays, and decision rules [17]. The modelling was developed using the software Vensim PLE 7.0 [18] and can be downloaded from Appendix A. The simulation unit was “day” and it used a long-term horizon of 730 days (two years). The Euler integration method was used with a time step of 0.25 by day, which means that on each simulated day, the software ran four times (1/0.25) for integration. The Euler integration method is a numerical technique used to solve ordinary differential equations (ODEs). It is a straightforward and widely used method due to its simplicity and ease of implementation. This method approximates the solution of ODEs by iteratively advancing the solution over small time steps [19].

Table 1 presents all the variables, while the equations used for model development are presented in Table 2.

The model diagram presents the established relationships between the variables and the feedback structure (Figure 1). The flow into the stocks can be written as Equations (5)–(7).

To validate the gain composition, REfat and REprot were divided by the energy content of fat (9.4) and protein (5.7), respectively, resulting in the rates *accfat* and *accprot*. These rates were utilized to accumulate the amount of nutrients retained during the simulation in the stocks of total body protein and total body fat. Using equations from Cannas et al. [16], an initial value for the fat and protein contents in relation to the initial body mass was then obtained. Environmental conditions were not added as input because this information is not usually included in research papers, so we consider the animals in thermoneutral condition.

### 2.2. Diet Composition

The diets used in the main simulation were from Sheridan et al. [21]. The composition of the low-energy and high-energy diets fed to Mutton Merino lambs are detailed in Table 3. These diets consist of a mixture of roughage, concentrated feed, vitamins, and minerals to meet the nutritional requirements of the animals.

### 2.3. Simulation Procedure

The simulation procedure was conducted in the following steps:Input Data: The initial input data consisted of the animal’s initial body mass (kg), standard final mass (kg), age (days), sex (1 or 1.15), and diet metabolizable energy concentration (Mcal.kg^−1^).Nutritional Requirements: The model calculated the nutritional requirements for maintenance (Mcal EM·day^−1^) and estimated the dry matter intake (kg·day^−1^). The metabolizable energy intake above maintenance (dimensionless) was then calculated, given the metabolizable energy available for growth (Mcal EM·day^−1^).Energy Content of Gain: Based on the metabolizable energy available for growth and the degree of maturity, the energy content of gain (Mcal·day^−1^) was estimated as the sum of the retained energy from protein (Mcal·day^−1^) and the retained energy from fat (Mcal·day^−1^). The protein in gain (g·day^−1^) and the fat in gain (g·day^−1^) were then calculated based on the retained energy from protein and fat, respectively. These values were accumulated in the total body protein (kg) and total body fat (kg).Net Energy on Gain: Based on the proportion of energy retained as protein, the *k_G_* was calculated and multiplied by the metabolizable energy to gain, giving the net energy on gain.Average Daily Gain: The average daily gain was estimated using the net energy available for growth. This calculation considers the initial body mass and its respective maintenance energy requirement. This change was automatically computed to the animal body mass, closing the feedback loop, and dynamically influencing the intake capacity and nutritional requirements over time.

The feedback structure of the simulation model is designed to adjust the animal nutritional requirements dynamically as it gains or loses mass, based on the energy available in the diet. This feature allows for the exploration of different growth scenarios by varying the energy concentration of hypothetical diets and the initial condition of the animal.

### 2.4. Model Evaluation as a Support Decision Tool

The model was validated using a dataset of independent studies to assess the ability of final body mass to predict outcomes. The selection criteria for the study included a range of factors such as initial and final body mass, dry matter intake, average daily gain, metabolizable energy or total digestible nutrients, breed or crossbreed, sex, initial age, and feedlot time. The dataset reference, number of observations, standard final mass, and genotype are described in Table 4, while Table 5 presents a summary of the statistics of the dataset. 

The standard final mass (SFM) for different breeds or crosses was determined based on available information from the “Breeds of Livestock” web page from Oklahoma State University and the Domestic Animal Genetic Resources Information System as described by Kemp et al. [22].

**Table 4 animals-14-02246-t004:** References, number of treatments, number of observations, standard final mass, and genotype of evaluation dataset.

Reference	Treatments	Observations	SFM ^a^ (kg)	Genotype
[23]	4	20	55	Pelibuey
[24]	3	18	100	Santa Inês
[21]	2	16	105	Mutton Merino
[25]	2	14	80	Awassi
[26]	3	54	81	Mehbraban
[27]	6	36	35	Taleshi
[28]	4	24	100	Santa Inês
[29]	4	24	100	Santa Inês
[30]	3	27	100	Santa Inês
[31]	1	10	105	Dorper × Katadhin
[32]	4	80	80	Chall
[33]	4	24	100	Santa Inês
[34]	5	45	100	Santa Inês
[35]	3	72	119	Hampshire
[36]	6	36	80	Chall
[37]	4	24	81	Mehbraban
[38]	4	40	75	Pelibuey × Katahdin

^a^ Standard Final Mass.

**Table 5 animals-14-02246-t005:** Dataset summary ^a^.

Item	Mean	Standard Deviation	Minimum	Maximum
Days on feed	68.85	18.99	35.00	105.00
Dietary ME (Mcal·day^−1^ of DM)	2.54	0.21	2.10	2.96
Initial body mass (kg)	25.85	5.90	18.05	39.10
Final body mass (kg)	39.94	8.87	29.15	61.70
Dry matter intake (kg·day^−1^)	1.30	0.30	0.87	2.33
Average daily gain (kg·day^−1^)	0.22	0.05	0.10	0.31

^a^ Dataset comprised 17 studies with 62 treatments for a total of 564 individual measures in growing male lambs.

Statistical evaluation was conducted between the simulated and observed values for final body mass. To evaluate the accuracy and precision of the model, the mean bias (Equation (24)), mean square error (MSE) (Equation (25)); root mean square error (RMSE) (Equation (27)), coefficient of determination (R^2^) (Equation (26)), and the MSE decomposition into mean bias (UM), systematic bias (US), and random error (UC) (Equations (29), (30), and (31), respectively) were used (Table 6).

To evaluate the model as a descriptive tool, the main variables are presented in a graphic form to describe their behaviors over time and their patterns compared to biological functions and scientific data. The simulation was conducted using the data from Sheridan et al. [21] as input; they worked with South African Mutton Merino lambs in feedlot receiving a high- and a low-energy diet of 2.89 and 2.36 Mcal ME·day^−1^, respectively. The input data was standard final mass (105 kg), sex (intact male), initial age (90 days), and initial mass of 33.31 kg for lambs fed with a high-energy diet and 32.13 kg for lambs fed with a low-energy diet.

## 3. Results

### Model Evaluation

The data used for model evaluation as diet energy concentration, breed or crossbred, age, and initial, final, and standard body mass are presented in Appendix A. The linear regression between the observed and simulated values for final body mass is presented in Figure 2. The model evaluation generates the following information: The mean bias (MB) was 0.656 kg, corresponding to 1.64% of the average final body mass of 39.94 kg;The mean square error of prediction (MSE) was 9.68;The root mean square error of prediction (RMSE) was 3.11 kg;The mean bias (UM), systematic bias (US), and random error (UC) values were 0.044, 0.024, and 0.93, respectively.

The Dynamic Lamb Growth Model is a dynamic tool that can provide a detailed description of nutrient requirements, composition of gain, and animal performance over time. By evaluating these parameters, it is possible to gain insights into the metabolic and physiological processes that underlie animal growth and development that could be used as a learning tool (Figure A1). 

Figure 3 shows the metabolizable energy intake (MEI), metabolizable energy intake above maintenance factor (L), metabolizable energy required for maintenance (MEm), metabolizable energy available for growth (MEg), metabolizable protein required for maintenance (MPm), and metabolizable protein required for growth (MPg). These parameters are essential for understanding the energy and protein requirements of animals, as well as their capacity for growth and production. The model outputs provide additional information of the maturity degree, energy content of gain (EVG), retained energy from protein (REprot) and fat (REfat), partial efficiency of ME to NE for growth (*k_G_*), and net energy to gain (NEg) over time. These parameters can help to identify the factors that influence animal growth and development, including the efficiency of nutrient utilization and the composition of the gain. 

The results of the descriptive evaluation show that the metabolizable energy intake (MEI) is influenced by changes in dry matter intake and differences in diet energy concentration. The MEI was 4.03 and 5.10 Mcal·day^−1^ for the first day of simulation and the fifty-sixth day in the low-energy group and 5.07 and 6.85 Mcal·day^−1^ for the first and fifty-sixth day of simulation in the high-energy group. In other words, the MEI was higher in the high-energy group compared to the low-energy group, with a difference that increased over time. Similarly, the metabolizable energy intake above the maintenance factor (L) was also higher in the high-energy group, indicating that animals in this group had more energy available for productive functions. The values for L from the first day were 2.09 and 2.45 times the metabolizable energy required to maintain the low- and high-energy feed levels, respectively.

The metabolizable energy required for maintenance (MEm) and metabolizable protein required for maintenance (MPm) increased with body mass, following a sigmoid pattern. Both the MEm and MPm were dependent on BM on their respective scales. The values of MEm were 1.93, 2.45, and 4.17 Mcal·day^−1^ on Day 1, 56, and 730, respectively, on a low-energy diet. The MPm was 52.31, 65.77 and 95.29 g·day^−1^ on Day 1, 56 and 730, respectively, on a low-energy diet. 

The metabolizable energy available for growth (MEg) was also higher in the high-energy group and increased over time, reflecting the greater energy available for productive functions in this group. The metabolizable protein required for growth (MPg) was influenced by the growth rate, with higher requirements in animals with higher rates of growth. The MEg was 2.10 and 3.00 Mcal·day^−1^ on the first day of the low- and high-energy feed, respectively, a difference of 0.90 Mcal·day^−1^. This difference increased on the fifty-sixth day, where the values were 2.65 and 4.01 Mcal·day^−1^, a difference of 1.36 Mcal·day^−1^. The MPg had a distinct behavior as a function of their dependency on the growth rate (ADG). On Day 1, 56 and 730 of the simulation, the simulated requirement for the MPg was 54.21, 72.39, and 1.04 g·day^−1^ at high-energy levels, respectively. 

Figure 4 presents the total body protein and total body fat as a function of body mass (BM) simulated at 56 days and 730 days in animals fed high-energy diets. The figure displays the patterns of protein and fat retention as a function of body mass, which can provide insight into the changes in nutrient partitioning that occur as animals grow and develop. 

Based on initial estimations, the total body protein was recorded at 5.08 kg, and the total body fat was recorded at 7.55 kg. As time progressed, the total body protein increased to 7.59 kg and 13.42 kg on the 56th and 730th days, respectively. This increase followed an exponential pattern that eventually stabilized at a standard final mass and maturity. On the other hand, the total body fat exhibited a different pattern of growth. The total body fat increased to 13.48 kg and 39.31 kg on the 56th and 730th days, respectively, following an exponential behavior pattern that showed a tendency to increase continuously over time.

The simulated patterns of the maturity degree, the energy content of gain (EVG), the energy retained as protein (REprot) and as fat (REfat), the partial efficiency of metabolizable energy to net energy (*k_G_*), and the net energy evaluable to gain (NEg) as a function of time were calculated. This demonstrated that the maturity degree (P) increased over time in a sigmoid pattern, approaching the standard final mass. As the simulation starts (day 0), the P value is 0.306 and 0.317 in lambs weighing 32.13 and 33.31 kg, respectively, and reaches 0.99 and 0.99 at 730 days, almost reaching the standard final mass. The energy content of gain (EVG) also increased over time, with a sigmoid pattern that reflected the changes in diet energy content. EVG ranged between 3.36 and 3.52 Mcal.kg^−1^ on day zero and between 3.96 and 4.34 Mcal.kg^−1^ on the fifty-sixth day of simulation on low- and high-energy diets. 

The retained energy from protein (REprot) decreased over time, while the retained energy from fat (REfat) increased, reflecting the changes in nutrient partitioning that occur during growth and development. REprot is assumed to be 0.908 and 0.929 Mcal.kg^−1^ on low-energy and high-energy diets, respectively, on the first day of simulation. REprot is reduced to 0.75 and 0.82 Mcal.kg^−1^ on the fifty-sixth day of simulation. REfat increases over time, being 2.44 and 2.62 Mcal.kg^−1^ on the first day of simulation on low- and high-energy diets, respectively, and 3.13 and 3.59 Mcal.kg^−1^ on the fifty-sixth day of simulation on low- and high-energy diets, respectively. 

The *k_G_* parameter was relatively consistent over time, ranging between 0.479 and 0.513 in animals between 32.13 kg and 43.51 kg during the fifty-six days that they were fed with the low-energy diet (2.36 Mcal ME.kg^−1^), and between 0.488 and 0.538 in animals between 33.31 kg and 51.43 kg during the fifty-six days that they were fed with the high-energy diet (2.89 Mcal ME.kg^−1^). The net energy available for gain (NEg) followed a pattern similar to the MEg multiplied by *k_G_*, with higher values in the high-energy group and at later stages of growth, assuming the values of 1.00 and 1.47 Mcal·day^−1^ on the first day and 1.37 and 2.16 Mcal·day^−1^ on the fifty-sixth day on low- and high-energy levels, respectively.

Finally, Figure 5 illustrates the relationships between dry matter intake (DMI), growth rate (ADG), and body mass (BM) change, which are critical factors in determining the energy and nutrient requirements of animals over time.

Regarding the simulated patterns of the dry matter intake (DMI), growth rate (ADG), and body mass (BM), dry matter intake behavior is closely related to the BM, expressing an ascending growth, and reaching a peak intake on the 263rd and 183rd days of simulation on low- and high-energy diets. The DMI started at 1.70 and 1.75 kg day^−1^ in the animals with an initial BM of 32.13 and 33.31 kg and reached 2.80 kg·day^−1^ in both categories after 730 days of following the diet. This corresponds to 5.29% and 5.25% of BM at the beginning and 2.66% of BM at the end of simulation. 

Finally, the growth rate (ADG) showed an expressive development in the animals fed with high-energy feed versus low-energy feed. The values verified from the growth rate simulation range from zero grams when the animal reaches the standard body mass to 0.277 and 0.341 kg·day^−1^ as the maximum values on low- and high-energy diets, respectively. The body mass is the accumulation of ADG over initial body mass, presenting the sigmoid pattern with the clear difference between animals fed with high-energy diets. 

## 4. Discussion

The regression analysis between observed and simulated values for final body mass presented significant parameters (*p* < 0.01) for the intercept and slope, demonstrating a strong correlation between the observed and simulated data. The R^2^ was relatively high, indicating precision. The variation in RMSE accounts for just 7.79% of the observed mean of the final body mass, while the unexplained variation (UC) accounts for 93%. This indicates that the model estimates the mean and the trend accurately, demonstrating both precision and reliability. The statistical assessment of the quality of simulation of the body mass at the end of the experimental period of the dataset studies shows precision and accuracy; thus, it indicates that the model can be used as a predictor of animal growth. 

The model provides a graphical and numeric description over time of all the variables involved, simulating the different processes and behaviors of interrelated variables. This graphical presentation and the easy handling changes in inputs can be useful as a learning tool; hence, it is possible to demonstrate any range of standard final mass, initial mass, and diet energy concentration and instantly visualize the nonlinear interactions and results.

In both diets used for a descriptive purpose, the energy is above maintenance (L), allowing growth. It can be seen that the factor reduces as the energy requirements of maintenance and the energy content of gain increase (Figure 5) and the gap between low and high levels of feed energy is discernible. In this model, the growth stops when the maturity mass reaches the imposed maturity degree, 105 kg in this example, which means that an overfeeding situation before reaching maturity mass cannot be accounted for by the model. In both diets, the animals are still receiving more energy than necessary for their maintenance.

The verified values of MEm and MPm are higher in lambs fed with high-energy diets, which is a result of cyclical changes in body mass, where high-energy-fed animals had a high body mass, which is used to account for the MEm and MPm. The metabolizable energy available for growth (MEg), which accounts for the difference between metabolizable energy intake and metabolizable energy for maintenance, displays the difference between the low and high levels of energy diets. It is lower at the beginning of the simulation when the body mass is closed and it later changes following the body mass change until it almost reaches the asymptote, being close to the standard final mass.

The metabolizable protein required for growth (MPg) is a direct function of average daily gain (ADG) (Figure 4). When the usual bell-shaped curve of the ADG is plotted over time, the animal reaches standard final mass, and the MPg tends to be zero [40]. A diet with high-energy content requires more MPg and low-energy diets require less MPg to support the estimated body mass change. 

The maturity degree or relative size (P) is used by CSIRO [41,42] and CSIRO-based systems to predict the protein, fat, and energy content of gain in cattle and sheep, as well to predict the potential dry matter intake. The verified differences between the *p* values imply a direct relationship with the composition of the gain, reflecting the higher energy content of the gain, the diminution of energy retained from protein, and the increase of energy retained from fat in animals fed with a high-energy diet. According to [43], it is expected that in young or less-developed animals, the proportion of retained protein is greater than in older or better-developed animals.

The equation for computing the partial efficiency of ME to NE (*k_G_*) is one differential of the SRNS model because the current models use empirical relationships as linear [41] or nonlinear models [20] and the SRNS equation works in a mechanistic way, using the proportion of energy retained as protein as a factor. The application of the equations proposed by the current models, $k_{G}=0.043\;ME$ (MJ kg^−1^ DM) [41] and $k_{G}=\left[\left(1.42\times ME-0.174\times {ME}^{2}+0.0122\times {ME}^{3}\right)-1.65\right]/ME$ (Mcal kg^−1^ DM) [20], give energy concentration 0.425 and 0.378 values of *k_G_* to the partial efficiency of ME to NE, respectively. Xu et al. [44] found a *k_G_* of 0.419 (±0.021) in lambs between 20.3 and 35 kg and Deng et al. [45] found a *k_G_* of 0.46 (±0.038) in lambs between 35 and 50 kg. Both sets of researchers compared the empirical estimates with the NRC [20] and CSIRO [41], and found underestimated *k_G_* values for both systems. 

The dry matter intake was highly influenced by the body mass and diet energy concentration; thus, as the animals increased their body mass, they also increased the dry matter intake. Hence, this causal relationship generates a pattern with a substantial increase at the beginning and a stabilization when the standard final mass is reached. The feed intake per unit of mass is predicted to be greater when the animal is young in comparison with mature animals, when the potential intake declines with increasing body condition at an advanced age [43,44,45].

The bell-shaped curve is commonly observed in the growth rate (ADG) when nonlinear functions are adjusted to empirical data [40,46,47,48]. The highest point of the curve is related to the inflection point of the sigmoid pattern of the growth curve. In the original research, the average reported daily gain is 0.203 and 0.281 g on low- and high-energy diets, respectively [21]. An interesting point of the simulation occurs on the 166th day on feeding, according to the simulation. After this point, animals that received low-energy feed had a higher growth rate than animals that received high-energy feed; at this point the body mass was 70.27 and 82.99 kg on low- and high-energy feed, respectively. This can be explained by the composition of gain. At this point, the animal that was fed high-energy feed was closer to reaching its maturity mass, and the adipose tissue deposition was theoretically higher than the muscle tissue deposition. One gram of protein has an energy content of about 5.7 kilocalories and one gram of fat has an energy content of about 9.4 kilocalories, so as animals grow to their advanced stages of maturity, they become less efficient in terms of body mass gain per unit of feed eaten [47].

No nutritional model establishes the inflection point in the growth of the animals, nor the point at which the animals reach maturity. This is because these models commonly aim to describe animal growth over a short period, without detailing the point where the variation in body mass becomes minimal, i.e., the body matures. The first model developed attempts to generate an exponential positive curve and not a standard sigmoid pattern, verified with empirical growth studies, where multiple nonlinear models are tested with goats and sheep [49,50,51]. The sigmoid pattern can be obtained in dynamic models accounting for the dependence of nonlinear interactions between positive and negative loops, where the positive loop initially promotes exponential growth up to a point where it loses its dominance to a negative loop and the system slowly tends to equilibrium, in our case, the body mass at maturity.

According to Brody and Lardy [52], animal growth is a transitory state in which feedback mechanisms (positive and negative, mediated by nutrients, hormones, and other signals) and the assimilation, transport, storage, or mobilization of nutrients combined, induce temporal delays that are plotted against time and resemble a growing sigmoid pattern. Body mass presents a sigmoid pattern that represents the accumulation of the growth rate as a stock. The inflection point occurs at the sixty-fifth and ninetieth days on feed with high and low energy levels, respectively. The final body mass simulated at the fifty-sixth day of feeding was 51.43 kg and 44.45 kg on feed with high and low energy levels, respectively, and the empirically observed data were 49.05 kg and 43.51 kg [21] on the same day, showing a slight overestimation by the simulated data.

## 5. Conclusions

The application of the Dynamic Lamb Growth Model provides a tool to simulate, analyze, and understand nutritional requirements, composition of gain, and lamb growth in feedlot systems. The model simulates the final body mass in a consistent manner, presenting both precision and accuracy. Changes in nutritional requirements and composition of gain over time were consistent with physiological and nutritional interaction knowledge, which was corroborated with the original reports. 

The dynamic model can be used as a support decision tool to estimate final body mass and the days of feed required to reach a certain final body mass, according to the different energy levels in each scenario. Moreover, the model can be used as a learning tool to illustrate the practical principles of animal nutrition, nutrient requirement relationships, and body composition changes, in addition to the understanding of standard body mass implications, where it is possible to simulate a wide range of breeds (standard body mass), initial mass, and diet energy concentration, understanding the dynamic interactions between variables and achieving performance simulations. 

## Figures and Tables

**Figure 1 animals-14-02246-f001:**
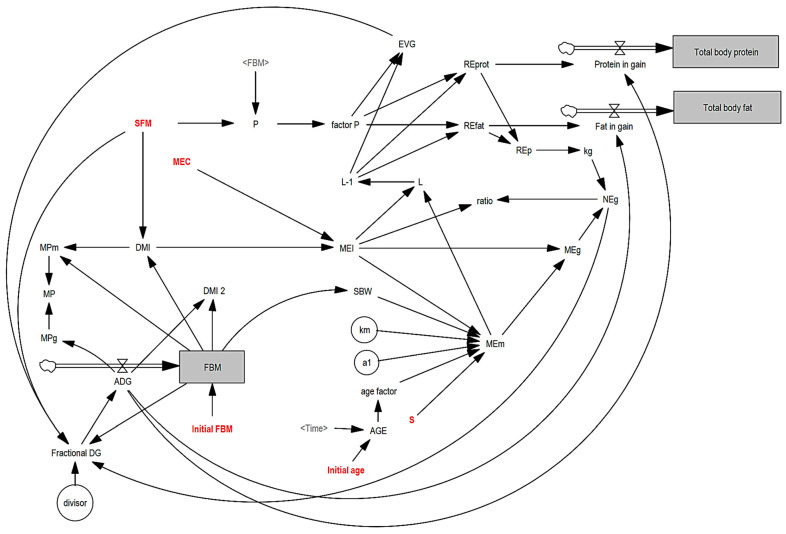
Causal relationship between original variables. Words and acronyms in red are user input, inside circles are constants, words in black are equations and relations, and below the thick arrow are the flow to gray boxes (stock). The thin arrows are the connections between the variables.

**Figure 2 animals-14-02246-f002:**
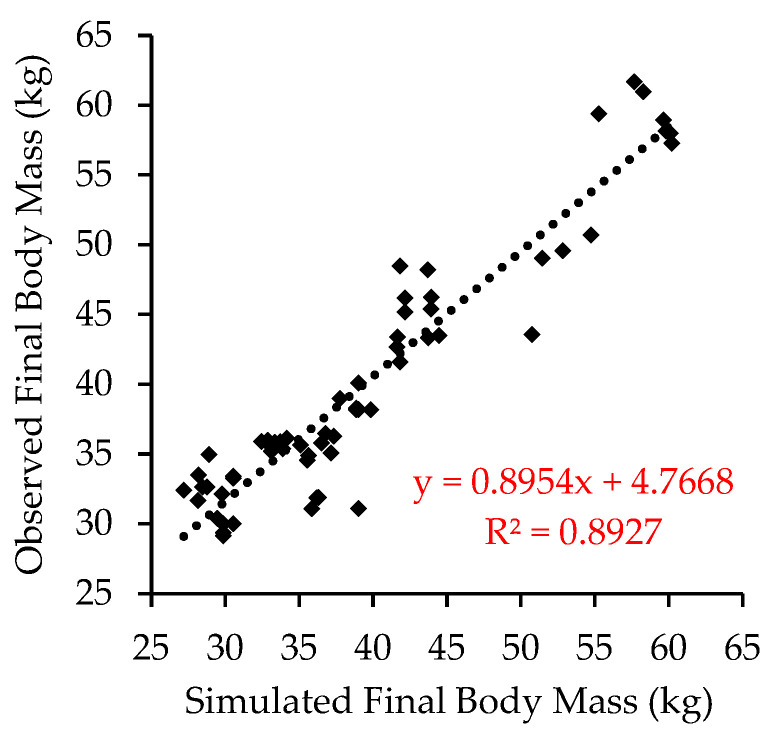
Regression of simulated against observed final body mass.

**Figure 3 animals-14-02246-f003:**
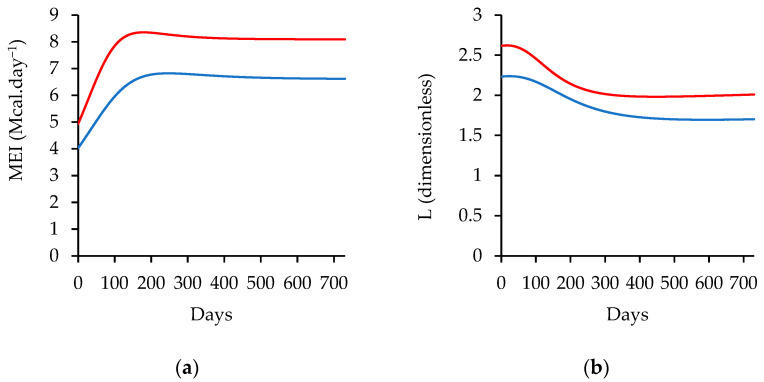
(**a**) Metabolizable energy intake (MEI), (**b**) metabolizable energy intake above maintenance factor (L), (**c**) metabolizable energy required for maintenance (Mem), metabolizable protein required for maintenance and MPm), (**d**) protein required for maintenance, (**e**) metabolizable energy available for growth (MEg), and (**f**) metabolizable protein required for growth (MPg) simulated with the input data from Sheridan et al. (2003) [21]. Blue line represents lambs fed low-energy diet (2.36 Mcal ME.kg^−1^), and the red line the lambs fed high-energy diets (2.89 Mcal ME.kg^−1^).

**Figure 4 animals-14-02246-f004:**
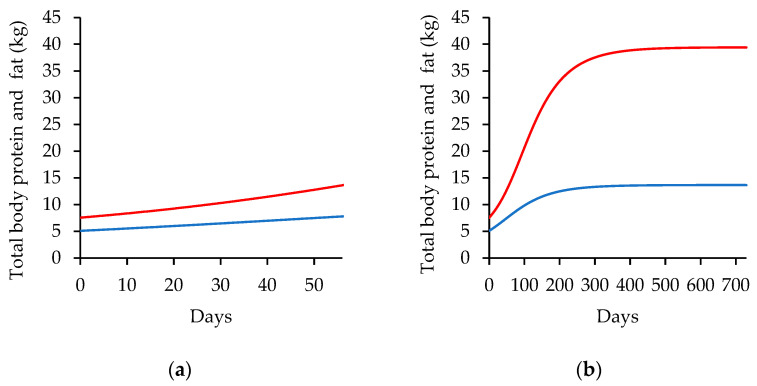
Total body protein (blue line) and total body fat (red line) in kilograms as function of body mass simulated with the input data from Sheridan et al. (2003) [21] for lambs fed high-energy diets (2.89 Mcal ME.kg^−1^) for 56 days (**a**) and 730 days (**b**).

**Figure 5 animals-14-02246-f005:**
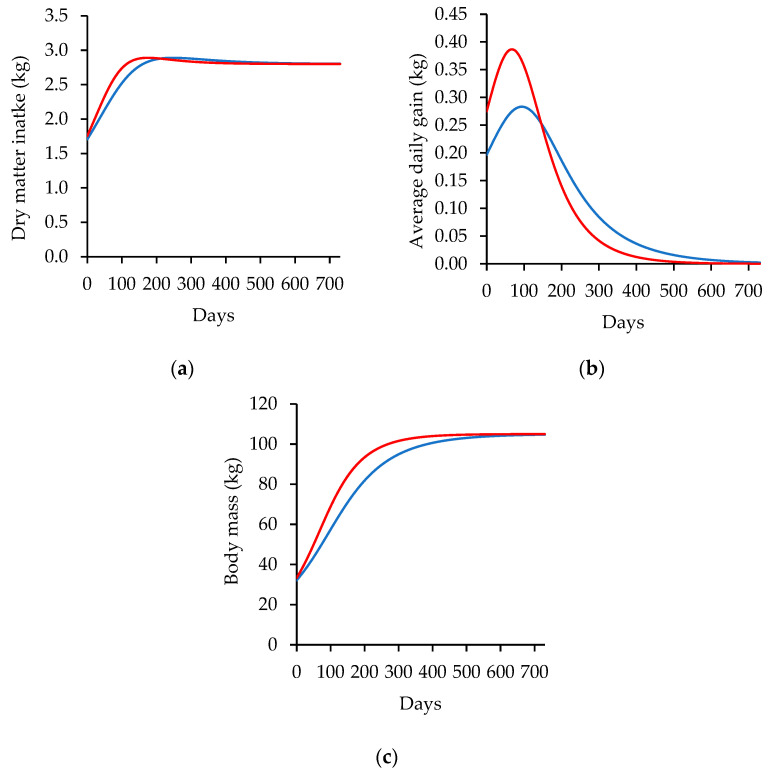
(**a**) Dry matter intake (DMI), (**b**) growth rate (ADG), and (**c**) body mass (BM) simulated with the input data from Sheridan et al. (2003) [21] for 730 days. Blue lines represent lambs fed low-energy diets (2.36 Mcal.ME kg^−1^), and the red line represents the lambs fed high-energy diets (2.89 Mcal ME.kg^−1^).

**Table 1 animals-14-02246-t001:** Description of the units and abbreviations of variables used in the model development and evaluation.

Variable	Unit	Description
$\bar{Y}$	^a^	Average simulated value
$Y_{i}$	^a^	*i*th observed or measured value
$s_{Y}^{\;}$	dimensionless	Standard deviation of the Y variable
*a*1	NE_M_ kg ^0.75^	Maintenance requirement
ADG	g·day^−1^	Average daily gain
AF	%	Proportion of empty body fat
Age	Days	Current age
AP	%	Proportion of empty body protein
BCS	1 to 5	Body condition score
BM	kg	Body mass
CP	g·day^−1^	Crude protein
DBF_(t)_	kg	Dynamic body fat
DBM_(t)_	kg	Dynamic body mass
DBP_(t)_	kg	Dynamic body protein
DMI	kg·day^−1^	Dry matter intake
EBM	kg	Empty body mass
EVG	Mcal·day^−1^	Energy content of gain
*k* _G_	dimensionless	Partial efficiency of ME to NE for growth
*k* _M_	dimensionless	Partial efficiency of ME to NE for maintenance
*L*	dimensionless	Intake above maintenance
MB	dimensionless	Mean bias
ME	Mcal.kg^−1^	Metabolizable energy
ME_G_	Mcal·day^−1^	ME available for growth
MEI	Mcal·day^−1^	ME intake
ME_M_	Mcal·day^−1^	ME requirement for maintenance
MP	g·day^−1^	Metabolizable protein
MP_M_	g·day^−1^	MP for maintenance
MSE	^a^	Mean square error of prediction
NE_G_	Mcal·day^−1^	Net energy available for growth
*P*	dimensionless	Degree of maturity
*r*	dimensionless	Pearson correlation coefficient.
R^2^	dimensionless	Coefficient of determination
RE_Fat_	Mcal·day^−1^	Retained energy from fat
RE_P_	dimensionless	Proportion of retained energy as protein
RE_prot_	Mcal·day^−1^	Retained energy from protein
RMSE	%	Root mean square error of prediction
S	dimensionless	Factor. 1 to female and castrated male and 1.15 to intact males
SBM	kg	Shrunk body mass
SBM^0.75^	kg	Metabolic SBM
SFM	kg	Standard final mass
*t* _0_	Days	Initial time
TE	Mcal	Total energy
TF	kg	Total body fat
*t_i_*	Days	Time in *i* moment
TP	kg	Total body protein
U^C^	dimensionless	MSE decomposition into random error
U^M^	dimensionless	MSE decomposition into mean bias
U^S^	dimensionless	MSE decomposition into systematic bias
n	unit	Sample size

^a^ = linked to the variable used.

**Table 2 animals-14-02246-t002:** Equations used in the model development [11,16].

Variable	Equation	Equation Number
ADG	$\left({NE}_{G}\times 1000)÷(0.92\times EVG\right)$	(1)
AF	0.0269+0.0869×BCS	(2)
AP	$-0.0039\times {BCS}^{2}+0.0279\times BCS+0.1449$	(3)
DMI ^a^	$6.8\times (BM/SFM)-4\times {\left(BM/SFM\right)}^{2}$	(4)
DBM_(t)_ ^b^	$\int_{t0}^{t}{ADG}_{\left(t_{i}\right)}ds+{BM}_{{(t}_{0})}$	(5)
DBF_(t)_ ^b^	$\int_{t0}^{t}{\mathrm{fat}\mathrm{in}\mathrm{gain}}_{\left(t_{i}\right)}ds+{\mathrm{Total}\mathrm{body}\mathrm{fat}}_{\left(t_{0}\right)}$	(6)
DBP_(t)_ ^b^	$\int_{t0}^{t}{\mathrm{protein}\mathrm{in}\mathrm{gain}}_{\left(t_{i}\right)}ds+{\mathrm{Total}\mathrm{body}\mathrm{protein}}_{{(t}_{0})}$	(7)
EBM	0.851×SBM	(8)
EVG	$0.239\times (6.7+2\times \left(L-1\right)+\left(16.5-2\times \left(L-1\right)\right)÷\left(1+e^{-6\times \left(P-0.4\right)}\right))\times 5.7$	(9)
k_G_	$18.36÷(27+41\times {RE}_{P})$	(10)
ME_M_ ^c^	$\left({SBM}^{0.75}\times a1\times S\times e^{\left(-0.03\times AGE\right)}+0.09\times MEI\right)÷k_{M}$	(11)
MP_M_	(0.147×BM+3.375)÷0.67+(0.1125×DMI)÷0.67	(12)
NE_G_	${ME}_{G}\times k_{G}$	(13)
RE_Fat_	$(43-56\times \left(L-1\right)-\left(\left(490-56\times \left(L-1\right))÷1+e^{-6\times \left(P-0.4\right)}\right)\right)\times 9.4$	(14)
RE_P_	${RE}_{Prot}÷({RE}_{Prot}+{RE}_{Fat})$	(15)
RE_Prot_	($212-8\times \left(L-1\right)-\left(\left(120-8\times \left(L-1\right))÷1+e^{-6\times \left(P-0.4\right)}\right)\right)\times 5.7$	(16)
SBM	BM×0.96	(17)
TE	9.4×TF+5.7×TP	(18)
TF	AF×EBM	(19)
TP	AP×EBM	(20)

^a^ Equation from NRC, 2007 [20], ^b^ Mathematical description of the flow into the stocks, and ^c^ Modified equation.

**Table 3 animals-14-02246-t003:** Ingredients and chemical composition of the diets simulated in this study, adapted from the values provided by Sheridan et al. [21].

Item/Ingredient	Low-Energy Diet (%)	High-Energy Diet (%)
Wheat Bran	1.50	0.00
Maize Meal	30.00	38.00
Sunflower Oilcake	1.64	4.46
Groundnut Oilcake	3.33	0.00
Limestone	0.07	0.58
Urea	0.70	0.59
Maize Germ	0.00	9.31
Supermax Premix	0.00	10.45
Vitamin and Minerals Premix	0.21	0.21
Monocalcium Phosphate	0.00	0.11
Salt	0.50	0.50
Citrus Ruman Flavor	0.02	0.02
Ammonium Chloride	0.75	0.75
Taurotec	0.03	0.03
NaOH Wheat Straw	21.26	15.00
Lucerne Hay	35.00	15.00
Molasses Cuber	5.00	5.00
Chemical Composition	Content	Content
Ash (%)	9.49	8.44
Crude Fat (%)	2.18	5.59
Crude Protein (%)	14.29	14.56
Acid Detergent Fiber (%)	24.74	17.78
Neutral Detergent Fiber (%)	44.17	35.92
Gross Energy (Mcal kg^−1^)	3.82	4.00
Metabolizable Energy (Mcal kg^−1^)	2.36	2.89
Calcium (%)	0.44	0.74
Phosphorus (%)	0.27	0.36

**Table 6 animals-14-02246-t006:** Equations used on model evaluation [17,39].

Variable	Equation	Equation Number
(Co) variance between Y and $f(X_{1}.\ldots .X_{p}{)}_{i}$	$s_{f\left(X_{1}.\ldots .X_{p}\right)Y}$	(21)
Average simulated value	$\bar{f}\;\left(X_{1}.\ldots .X_{p}\right)$	(22)
ith simulated value	$f(X_{1}.\ldots .X_{p}{)}_{i}$	(23)
MB	$\frac{\sum_{i=1}^{n}\left(Y_{i}-f(X_{1}.\ldots .X_{p}{)}_{i}\right)}{n}$	(24)
MSE	$\frac{\sum_{i=1}^{n}(Y_{i}-f(X_{1}.\ldots .X_{p}{)}_{i}{)}^{2}}{n}$	(25)
R^2^	${\left(\frac{s_{f\left(X_{1}.\ldots .X_{p}\right)Y}}{s_{Y}{\times s}_{f\left(X_{1}.\ldots .X_{p}\right)}}\right)}^{2}$	(26)
RMSE	$\sqrt{MSE}$	(27)
Standard deviation for $f{\left(X_{1}.\ldots .X_{p}\right)}_{i}$	$s_{f\left(X_{1}.\ldots .X_{p}\right)}$	(28)
U^C^	$\frac{2\left(1-r\right)s_{f\left(X_{1}.\ldots .X_{p}\right)}s_{Y}^{\;}}{MSE}$	(29)
U^M^	$\frac{\bar{f}\;\left(X_{1}.\ldots .X_{p}\right)-\bar{Y}}{MSE}$	(30)
U^S^	$\frac{s_{f\left(X_{1}.\ldots .X_{p}\right)}-s_{Y}^{\;}\;}{MSE}$	(31)

## Data Availability

The original contributions presented in the study are included in the article/Appendix A; further inquiries can be directed to the corresponding authors.

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
