# Peer review of "A Dynamic Tool to Describe Lamb Growth and Its Use as a Decision Support System"

_animals, 2024, doi:10.3390/ani14152246_

Round 1
Reviewer 1 Report
Comments and Suggestions for Authors
A manuscript submitted for Recession on "A Dynamic Tool to Describe Lamb Growth and Its Use as a Decision Support System" is relevant and interesting for science and practice as it examines the possibility of developing a growth management system in lambs. In this aspect, I welcome the idea of ​​developing this manuscript and presenting it in a scientific experiment.
In the manuscript I do not find some basic points that concern the breeding of all productive animals for meat such as:
What is the breed and gender of the lambs used in the experiment?
What is the composition of the ration - roughage, concentrated feed, vitamins and minerals?
What is the feed conversion in the trial and how will the introduction of this system contribute to the economic effect of rearing these lambs?
Author Response
Comments 1. In this aspect, I welcome the idea of ​​developing this manuscript and presenting it in a scientific experiment.
Response 1. Thank you for your consideration. This paper aims to present a dynamic model designed to simulate various scenarios without the need for animal research. Dynamic modeling has been utilized across multiple fields of study to replace certain experimental methods. While it does introduce a degree of error, it effectively describes behaviors and trends rather than specific data points.
Comments 2. In the manuscript I do not find some basic points that concern the breeding of all productive animals for meat such as:
Comments 2.1. What is the breed and gender of the lambs used in the experiment?
Response 2.1. The required information is clearly important and is detailed in the Table A. (Please see the attached Evaluation dataset).
Comments 3. What is the composition of the ration - roughage, concentrated feed, vitamins, and minerals?
Response 3. A summary of the treatments for each experiment is presented in Table A. (Please see the attached Evaluation dataset). The newly added Table 3 shows the diet composition from Sheridan et al. (2003), as it was used for the experiment simulated in the graphs with low and high energy concentration.
Comments 4. What is the feed conversion in the trial and how will the introduction of this system contribute to the economic effect of rearing these lambs?
Response 4. Feed conversion was not calculated. However, the model can estimate the total feed required and the time needed to reach the desired final body mass across various scenarios of initial body mass and diet compositions, aiding in decision-making.
Reviewer 2 Report
Comments and Suggestions for Authors
Including the Vensim PLE software version
Introduction
Clearly articulate how the proposed model addresses the current gap in the literature, emphasizing the limitations of existing static models and how a dynamic model can offer advantages in terms of accuracy and applicability.
Providing a brief explanation of dynamic models in the introduction can better contextualize the study by explaining how these models differ from static models and their relevance in forecasting and optimizing complex processes such as animal growth.
Consider a concise review of existing models in the literature, such as CNCPS, SRNS, and RNS, highlighting their approaches and specific limitations, to underscore how the proposed model represents a significant advancement.
Include details on the ordinary differential equations used and the theoretical foundations of the model.
A more detailed explanation of the Euler integration method and time step could justify the choice of this specific method and explain the impact of the time step on simulations.
It was noted that environmental conditions were not included, which could be a significant limitation of the model since environmental variables can profoundly influence lamb growth and nutrition.
What percentage does the mean bias of 0.656 represent?
Figure 2 does not display the coefficient of determination.
A work has scientific merit, however, it needs improvement, and the validation of the model with other data is essential.
Comments on the Quality of English LanguageTo conduct a brief review of the article
Author Response
Comments 1. Providing a brief explanation of dynamic models in the introduction can better contextualize the study by explaining how these models differ from static models and their relevance in forecasting and optimizing complex processes such as animal growth.
Response 1. The introduction was updated to include the information required in the last paragraph. Line 69 - Several models have been developed to improve our understanding of ruminant nutrition and growth. Notable among these are the Cornell Net Carbohydrate and Protein System (CNCPS) [11], the Small Ruminant Nutrition System (SRNS) [12], and the Ruminant Nutrition System (RNS), as well as NASEM [13,14]. The CNCPS is a mechanistic model that predicts the nutrient requirements and feed utilization for cattle based on rumen fermentation, microbial growth, and animal physiology. The SRNS, which evolved from the CNCPS, is tailored for sheep and goats, incorporating specific adjustments for small ruminants. Despite its advancements, it faces challenges in accurately predicting nutrient requirements for different breeds and environmental conditions. These models, with a mechanistic structure, establish these relations and give a significant contribution to their understanding with better descriptions of the different nutritional processes. All these models are static; they support only punctual estimation related to the animal needs and their individual performances without feedback or continuity on information flow. Line 82 - The RNS represents an advancement by integrating additional submodels and stochastic modeling capabilities to account for variability in feed composition and animal responses. It offers more accurate predictions by considering a wider range of variables and interactions, yet it still requires refinement to address the complexities of ruminant nutrition fully. A summary of these models is described by Tedeschi and Menendez (REF). Within these lines, this article introduces a dynamic model that aims to integrate equations of CNCPS-S, SRNS, and RNS models to obtain protein and energy requirements, body composition of gain and animal growth, and accounting cyclical changes over time on feedlot lambs before weaned until their maturity, using animal and diet characteristics as input.
Comments 2. Consider a concise review of existing models in the literature, such as CNCPS, SRNS, and RNS, highlighting their approaches and specific limitations, to underscore how the proposed model represents a significant advancement.
Response 2. Thank you for your suggestion, but we would like to refrain from adding a review of such models. This information is available in the literature, and we cited the comprehensive book chapter by Tedeschi and Menendez (2022), to sustain the statement in the text.
Comments 3. Include details on the ordinary differential equations used and the theoretical foundations of the model.
Comments 3.1. A more detailed explanation of the Euler integration method and time step could justify the choice of this specific method and explain the impact of the time step on simulations.
Response 3.1. We added the following information to the methodology: Line 100 The Euler integration method was used with a time step of 0.25 by day, which means that each simulated day the software ran four times (1/0.25) for integration. The Euler integration method is a numerical technique used to solve ordinary differential equations (ODEs). It is a straightforward and widely used method due to its simplicity and ease of implementation. This method approximates the solution of ODEs by iteratively advancing the solution over small time steps (Thornley and France, 2007).
Comments 4. It was noted that environmental conditions were not included, which could be a significant limitation of the model since environmental variables can profoundly influence lamb growth and nutrition.
Response 4. I completely agree with your observation. Unfortunately, the papers used on model evaluation, and most of the available literature, do not include environmental information. Without this data for model input, it is impossible to verify the model's precision and accuracy, potentially leading to unnecessary errors in the predictions.
Comments 5. What percentage does the mean bias of 0.656 represent?
Response 5. The mean bias result corresponds to 1.64 % of the average final body mass of 39.94 kg, information (included on text).
Comments 6. Figure 2 does not display the coefficient of determination.
Response 6. The coefficient of determination was inserted in Figure 2.
Comments 7. A work has scientific merit, however, it needs improvement, and the validation of the model with other data is essential.
Response 7. Thank you for your insightful comment and for recognizing the scientific merit of our work. We agree that validating the model is crucial for demonstrating its robustness and generalizability. To address this concern, we have included the raw data used for model evaluation as supplementary material (please see the attachment). This dataset includes information on 564 animals, under 62 treatments from 17 different studies. These datasets encompass various breeds, crossbreeds, initial and final weight, and diets, providing a comprehensive assessment of the model's performance across different scenarios.

Round 2
Reviewer 2 Report
Comments and Suggestions for Authors
In the evaluated version, I have just a few more suggestions.
In the results, it was stated, 'The linear regression between observed and simulated values to final body mass is presented in Figure 2.' I suggest including the equation with its parameters in the regression graph.
In the regression model, I recommend testing the hypotheses of the model parameters (significance) for validation and discussing them in the text.
In the summary, it was mentioned that the correlation coefficient was 0.8927, the same value presented for the coefficient of determination, which is not statistically possible unless the correlation is exactly 1 or 0. Please review the values to correctly identify the correlation and the coefficient of determination.
Author Response
Comments 1: In the results, it was stated, 'The linear regression between observed and simulated values to final body mass is presented in Figure 2.' I suggest including the equation with its parameters in the regression graph.
Response 1: Equation included in Figure 2.
Comments 2: In the regression model, I recommend testing the hypotheses of the model parameters (significance) for validation and discussing them in the text.
Response 2: Thank you for the comment. Both parameters were significant, and the information have been included in the first sentence of the Discussion section (lines 322, 323, and 324).
Comments 3: In the summary, it was mentioned that the correlation coefficient was 0.8927, the same value presented for the coefficient of determination, which is not statistically possible unless the correlation is exactly 1 or 0. Please review the values to correctly identify the correlation and the coefficient of determination.
Response 3: Thank you for your observation. It was my error during the translation process. Changes can be verified in lines 14 and 27.